# Severe Aortic Stenosis and Pre-Excitation Syndrome in Pregnancy—A Multidisciplinary Approach

**DOI:** 10.3390/diagnostics15162099

**Published:** 2025-08-20

**Authors:** Miruna Florina Ştefan, Lucia Ştefania Magda, Catalin Gabriel Herghelegiu, Doru Herghelegiu, Oana Aurelia Zimnicaru, Catalin Constantin Badiu, Maria Claudia Berenice Suran, Andreea Elena Velcea, Calin Siliste, Dragoș Vinereanu

**Affiliations:** 1Department of Cardiology and Cardiovascular Surgery, University and Emergency Hospital of Bucharest, 050098 Bucharest, Romania; stefan_miruna@yahoo.com (M.F.Ş.); catalin.badiu@umfcd.ro (C.C.B.); berenice.suran@umfcd.ro (M.C.B.S.); andreea.velcea@umfcd.ro (A.E.V.); calin.siliste@umfcd.ro (C.S.); vinereanu@gmail.com (D.V.); 2Department of Cardiology and Cardiovascular Surgery, Faculty of Medicine, University of Medicine and Pharmacy Carol Davila Bucharest, 020021 Bucharest, Romania; 3Sanador Clinical Hospital, 011021 Bucharest, Romania; herghelegiu.cata@gmail.com (C.G.H.); dr.herghelegiu@gmail.com (D.H.)

**Keywords:** pregnancy and cardiovascular disease, maternal–fetal medicine, aortic stenosis, pre-excitation syndrome, bicuspid aortic valve, pre-eclampsia

## Abstract

**Background/Objectives**: Heart disease affects 0.1% to 4% of pregnant women, with congenital heart defects being the leading cause in developed countries. While maternal mortality is generally low, pre-existing cardiac conditions substantially increase adverse outcome risks. This report describes the multidisciplinary management of a pregnant patient with a bicuspid aortic valve, severe aortic stenosis, and ascending aortic ectasia. **Case Presentation**: A 34-year-old pregnant woman, asymptomatic but at high risk (World Health Organization Class III) for hemodynamic decompensation, was closely monitored throughout gestation. At 36 weeks, intrauterine growth restriction was detected, prompting an elective cesarean delivery at 38 weeks. Postpartum, the patient developed pre-eclampsia, which was managed successfully. Imaging revealed progressive aortic dilation, leading to surgical aortic valve replacement and ascending aorta reduction plasty. Post-operatively, atrioventricular reentrant tachycardia from an unrecognized accessory pathway developed; medical therapy effectively controlled the arrhythmia after failed catheter ablation. One year later, both mother and child remained in good health. **Discussion:** This case illustrates the complexity of managing pregnancy in women with congenital heart disease and significant aortic pathology. The physiological changes of pregnancy can exacerbate underlying lesions, necessitating individualized risk assessment, vigilant monitoring, and timely intervention. **Conclusions**: A multidisciplinary approach involving cardiology, obstetrics, anesthesiology, and genetics is essential to optimize outcomes for pregnant women with significant heart disease. As advances in care allow more women with congenital heart defects to reach childbearing age, structured care pathways remain vital for ensuring safe pregnancies and long-term cardiovascular health.

## 1. Introduction

The prevalence of heart disease among pregnant women ranges from 0.1% to 4%. In developed nations, congenital heart defects are the primary cause. In these countries, the overall risk of death during pregnancy is very low. However, cardiac conditions remain the primary cause of maternal mortality [1]. Consequently, women with pre-existing heart disease face a mortality rate that is 100 times higher than those without such conditions [2]. Throughout pregnancy, hormonal fluctuations induce significant hemodynamic shifts, potentially triggering decompensation in previously asymptomatic cases. Bicuspid aortic valve (BAV) is the most common cause of aortic stenosis among women of childbearing age, accounting for 2% of congenital heart diseases [3,4,5,6,7]. Obstructive heart lesions, like aortic stenosis (AS), worsen during pregnancy due to increased stroke volume. Although AS is rare in women of childbearing age, its presence is linked to higher risks of maternal cardiovascular events—including death—as well as obstetric complications, such as preterm birth, and fetal issues, like growth restriction, miscarriage, stillbirth/as preterm birth, miscarriage, and stillbirth, as well as fetal complications, like growth restriction [8].

Pre-eclampsia is a serious pregnancy complication characterized by hypertension developing after 20 weeks of gestation, accompanied by proteinuria or other signs of organ dysfunction. It remains a leading cause of maternal and perinatal morbidity and mortality, especially in cases of early onset [9]. Despite extensive research aimed at identifying preventive strategies, its incidence has remained largely unchanged over the past few decades. Affecting 2% to 8% of pregnancies, pre-eclampsia contributes significantly to adverse maternal and neonatal outcomes and is responsible for one-sixth of all preterm births, placing a considerable burden on healthcare systems [10]. Postpartum pre-eclampsia is a more rarely encountered entity, that may pose a greater risk of maternal morbidity than its antepartum counterpart, yet it remains a largely understudied condition [11].

The unmasking of an accessory pathway following heart surgery is a rare phenomenon with limited documentation in the literature, and its underlying pathophysiology remains a topic of debate [12].

We present a case that highlights a unique combination of pregnancy in a patient with a BAV, severe stenosis, and ascending aorta dilation, which remained asymptomatic but led to intrauterine growth restriction and postpartum pre-eclampsia. Following aortic valve replacement, an accessory pathway was discovered. Despite these complications, successful management was achieved. The case underscores the importance of close multidisciplinary monitoring for pregnancies involving aortic valve pathologies and ascending aorta dilation.

## 2. Detailed Case Description

**a.** 
**Patient presentation**


A 34-year-old patient, 22 weeks pregnant, known to have aortic bicuspid aortic disease since childhood and diagnosed with moderate aortic stenosis and mild regurgitation 3 years priorly, was referred by her obstetrician for cardiological evaluation. She did not undergo a cardiac evaluation immediately before becoming pregnant. At presentation, she was asymptomatic and had good functional capacity. She had no other known pathologies. She took no medication at home, except for a prophylactic dose of aspirin that was prescribed by the obstetrician for the prevention of intrauterine growth restriction. The clinical examination showed a good general state, no signs of pulmonary or systemic congestion, oxygen saturation of 99%, normal blood pressures, symmetric between the superior limbs, regular heart sounds and a proto-mezo-systolic heart murmur in the aortic area. Exercise testing before pregnancy was not performed. We had an in-depth discussion with her, explaining that she met the criteria for modified World Health Organization (WHO) Risk Class III and faced a very high risk of adverse maternal and fetal outcomes. We thoroughly reviewed her options, including pregnancy termination, which she refused.

**b.** 
**Initial work-up**


Transthoracic cardiac ultrasound revealed preserved left ventricular systolic function (LVEF, left ventricular ejection fraction 59% calculated by Simpson biplane), with correct segmental kinetics, concentric left ventricular hypertrophy (13 mm), undilated left ventricle, undilated right heart chambers with preserved right ventricular longitudinal function, bicuspid aortic valve, intensely calcified, predominantly posterior shelf, with severe aortic stenosis (maximum aortic velocity 4.9 m/s, mean gradient 58 mmHg, indexed AVA determined, with the continuity equation 0.475 cm^2^/m^2^—see Table 1, Appendix A, Figure 1 and Figure 2), mild aortic regurgitation, dilated ascending aorta (45 mm), no other significant valvulopathies, undilated inferior vena cava, and normal pericardium. Aortic diameters were measured at the annulus, root, sino-tubular junction and ascending aorta, and the largest diameter was used. Measurements of the aorta were made using the end-diastolic leading edge-to-leading-edge convention.

The electrocardiogram showed sinus rhythm with antero-superior hemiblock and left ventricle hypertrophy elements, without ST-T changes. The PR interval was within normal range and no sign of pre-excitation was observed (Figure 3, Panel A).

The laboratory findings were within normal range, including a NT-proBNP of 45 pg/mL, except for a mild anemia that was attributed to the pregnancy. Considering the high maternal and fetal risk, multidisciplinary team follow-up was decided, in concordance with the recommendations of the ESC Guidelines in practice. Cardiological evaluation was planned to be monthly in the 2nd trimester and weekly in the 3rd trimester. The patient remained completely asymptomatic with good exercise capacity without echocardiographic or electrocardiographic changes (Table 1). Because of the maternal congenital heart disease, thorough fetal morphological serial screening by a specialist was performed. Antepartum evaluation of the fetus did not identify cardiac malformations (Figure 4). Therefore, screening for genetic mutations was deemed unnecessary.

**c.** 
**Diagnosis and Management**


The patient’s diagnosis was asymptomatic aortic disease (bicuspid valve with severe stenosis and mild regurgitation and ascending aorta ectasia). This classified her in class III of the Modified WHO classification of maternal cardiovascular risk, according to the 2018 ESC Guidelines for the management of cardiovascular diseases during pregnancy, with an increased risk of maternal mortality or severe morbidity.

After comprehensive discussions with the patient, her family, and the multidisciplinary team (cardiologist, obstetrician, anesthesiologist and cardiovascular surgeon), we chose to pursue vigilant outpatient monitoring, with no treatment except for the low dose aspirin prescribed by the obstetrician for pre-eclampsia prevention. Criteria for intervention would include occurrence and progression of heart failure symptoms, echocardiographic signs of declining cardiac function, or signs of fetal distress. The healthcare team thoroughly educated the patient about potential warning signs of worsening of her condition, such as shortness of breath, chest discomfort or syncope. The patient was encouraged to seek immediate medical attention at the specialized tertiary care center if she experienced any of these symptoms, ensuring prompt and expert care if her condition deteriorated. The patient and the baby were evaluated monthly in the second trimester and weekly in the third trimester of pregnancy. No statistically significant changes in the echocardiographic measurements were identified during monitoring. Additionally, NT-proBNP values did not increase, the electrocardiogram showed no changes and the patient remained asymptomatic, with good physical effort capacity and good results on 6 min walking test (6MWT) (see Table 1).

At 28 weeks, the patient was diagnosed with gestational diabetes following an oral glucose tolerance test. She was referred for a diabetology consultation, where dietary management without antidiabetic medication was recommended. Good blood glucose control was achieved and no changes in fetal growth were observed.

Although children of mothers with gestational diabetes are typically macrosomic, symmetrical intrauterine growth restriction was diagnosed at 36 weeks of pregnancy (growth chart showing flattening of the fetal growth curves, Doppler ultrasound showing fetal vascular redistribution with decreased resistance and pulsatility in the middle cerebral artery and an increased resistance and pulsatility in the umbilical artery) (Figure 5, Appendix A).

Given the dilation of the ascending aorta (47 mm), the risk of aortic dissection and cardiac function deterioration, and the emergence of fetal distress, an elective cesarean section was performed at 38 weeks of pregnancy. The procedure was carefully managed with specific anesthetic considerations, including general anesthesia to maintain blood pressure within a narrow range and have a better control of potential cardiovascular complications, such as pulmonary edema or cardiac arrest, continuous invasive monitoring of maternal hemodynamics and fetal status, and effective pain control. A multidisciplinary team was present in the operating room, with emergency cardiothoracic surgical services on standby. The surgery was completed without complications. The baby had a birth weight of 2550 g, a length of 45 cm and received an Apgar score of 9.

The patient developed postpartum pre-eclampsia within the first 48 h after delivery, with a maximum systolic blood pressure of 180 mmHg and confirmed proteinuria. The renal and hepatic functions and platelet count were within normal range. She was treated promptly with methyldopa, nifedipine, and metoprolol, achieving effective blood pressure control and maintaining a stable heart rate without any electrocardiographic abnormalities. The patient remained asymptomatic despite high blood pressure values.

At 6-month post-partum a thoracic aorta CT scan confirmed the presence of the BAV with dilatation of the ascending aorta up to 49 mm (Figure 6). A stress test was not performed (due to the recent cesarean section). The coronary angio-CT showed no atherosclerotic plaques. At this point, taking into account the above mentioned diagnosis but also the wish of the patient and in accordance with the recommendations of the 2021 ESC/EACTS Guidelines for the management of valvular heart disease (indication class IIa, level of evidence B) [13], surgical replacement of the aortic valve with ascending aorta reduction plasty was decided by the heart team. Successful aortic valve replacement with a mechanical prosthesis, chosen considering both the patient’s age and her stated wish to avoid future pregnancies, and supra-coronary non-reinforced reduction ascending aortoplasty (RAA) were performed via mini-sternotomy (with cardiopulmonary bypass time of 93 min, aortic cross-clamp time of 69 min, and estimated blood loss of 500 mL). A wedge-shaped segment of the ascending aorta was excised through a longitudinal aortotomy starting next to the aortic clamp at the greater curvature, and the aortotomy was closed with a continuous suture in a double-layered fashion. The decision to utilize a mechanical valve and RAA, rather than prosthetic replacement of the ascending aorta, was made in full agreement with the patient to ensure valve durability while avoiding more extensive surgery.

Echocardiography after surgery showed correct disc mobility, maximal aortic velocity of 2.3 m/s and mean gradient of 11 mmHg with and ascending aorta diameter of 35 mm (Figure 7, Appendix A).

**d.** 
**Follow-up**


On post-operative day 3, the patient had an episode of atrial fibrillation with rapid ventricular response and hemodynamic decompensation (dyspnea, systolic blood pressure 90 mmHg), converted to sinus rhythm by external electric shock. Beta-blockers, i.e., metoprolol succinate 100 mg/day, were prescribed as secondary prevention of the atrial arrhythmia. One month post-operatively, the patient developed several episodes of rapid paced palpitations, documented electrocardiographically as orthodromic Atrioventricular Re-entry Tachycardia (AVRT) (Figure 3, Panel B). On conversion to sinus rhythm, left posterior type delta wave was present (Figure 3, Panel C). The evidence of the delta wave post-operatively was thought to be the consequence of a slight slowing of the nodal atrio-ventricular conduction secondary to heart surgery. Cardiac ultrasound and laboratory parameters were not modified. Transcatheter ablation was attempted but, due to anatomical reasons, placement of a probe in the coronary sinus was not successful. The antiarrhythmic treatment was changed to a combination of 450 mg/day propafenone and bisoprolol. To date (two years after the last procedure), the patient did not repeat any arrhythmia and both she and her baby remain in good health.

## 3. Discussion

Managing pregnancy in women with severe aortic stenosis is challenging due to complex clinical presentations. As noted in the literature, the differential diagnosis for heart failure symptoms in pregnancy includes physiological changes, cardiomyopathies, endocarditis, myocarditis, pulmonary embolism, myocardial infarction, spontaneous coronary artery dissection, arrhythmias, and pulmonary hypertension [14]. Pregnancy induces marked cardiovascular changes: cardiac output rises by 30–50%, initially due to a 40% increase in plasma volume and later due to a 15–20% rise in heart rate, before declining slightly in late pregnancy due to uterine compression. During labor, cardiac output rises further—by 15–20% early, 50% in active labor, and up to 80% postpartum—driven by pain, contractions, and autotransfusion after delivery [15,16,17]. Systemic vascular resistance and mean arterial pressure fall early due to placental circulation and vasodilators. Relative plasma volume expansion leads to physiologic anemia, reducing viscosity and improving placental perfusion. Most changes resolve within two weeks postpartum, with complete recovery in about six months, but they can significantly increase risks in women with cardiac disease [18]. A severe, fixed stenosis, whether aortic or mitral, can severely restrict forward blood flow and hinder the necessary increase in cardiac output, potentially resulting in congestive heart failure, pulmonary edema, shock, and even death [19]. Moreover, the increase in blood volume that occurs during pregnancy, as well as the compression of the inferior vena cava by the uterus, may lead to cardiac decompensation [20]. Among the most frequent adverse fetal outcomes reported are preterm delivery and intrauterine growth restriction [8]. In our case, i.e., the presence of a fixed outflow obstruction, the capacity to further augment stroke volume is limited, which may impair uteroplacental perfusion even without an overt low-output state. This relative hemodynamic restriction can lead to chronic placental hypoxia and intrauterine growth restriction, as confirmed by Doppler ultrasound assessment (cerebral vasodilatation—brain sparing and umbilical cord artery vasoconstriction were clear signs of fetal distress, while the uterine arteries had normal Doppler waveforms, meaning that trophoblast invasion and uterine artery remodeling were normal and not the cause of placental hypoperfusion).

Pregnancy poses an extremely high risk of maternal morbidity and mortality in certain severe cardiac conditions, classified as modified WHO-IV. In such cases, continuing the pregnancy is strongly discouraged, and termination is advised if conception occurs. One of these conditions is AS, which becomes particularly dangerous when accompanied by impaired left ventricular systolic function (ejection fraction < 50%) or an abnormal exercise stress test. Pregnancy is also contraindicated in women with an isolated BAV if the ascending aorta is dilated beyond 50 mm. For those with a BAV associated with Marfan syndrome or other hereditary thoracic aortic diseases (HTADs), the threshold for risk is lower, with ascending aortic dilation exceeding 45 mm being a critical concern [1].

In addition, according to the 2020 ACC/AHA guidelines, women with AS who are considering pregnancy should undergo valve intervention, even if asymptomatic [21]. This proactive approach aims to reduce potential risks associated with the increased cardiovascular demands of pregnancy and childbirth in patients with severe AS.

In asymptomatic patients, which was the case of our patient, extended preconception assessment, including an exercise test, should be performed [1]. This was unfortunately not possible for our case, since the patient firstly presented for a cardiological examination when she was already 22 weeks pregnant. An abnormal exercise test is defined as the following exercise-induced changes: development of symptoms, abnormal ECG, failure of improvement of left ventricular ejection fraction, failure of blood pressure rise or blood pressure drop during exercise, exercise-induced rise in the mean aortic gradient >20 mmHg or exercise-induced systolic pulmonary hypertension >60 mmHg [22].

When monitoring a pregnant woman with AS, it is important to keep in mind that the hemodynamic changes induced by pregnancy may increase flow-dependent measurements, like peak velocity and transvalvular pressure gradients, by approximately 50%. However, the aortic valve area typically remains unchanged throughout gestation [23].

The ROPAC registry (Registry of Pregnancy and Cardiac Disease) has gathered valuable information regarding the risk of pregnancy in moderate and severe aortic stenosis. Hospitalizations were quite common, with 35.8% of patients experiencing at least one admission during pregnancy, 20.8% of these admissions being for cardiac pathology, and the rate increased with the severity of the aortic valve lesion. Overall, hospitalizations for any reason occurred at a mean gestational age of 26.9 weeks. Heart failure affected 6.7% of previously asymptomatic patients but rose to 26.3% in symptomatic women with severe AS. Cesarean section rates were higher in severe AS (75.0%) versus moderate AS (48.3%), mostly for maternal indications (65.9%). Neonates of mothers with severe AS had lower median birth weights (3000 g vs. 3200 g) and shorter gestation (37.9 vs. 39.0 weeks). Peak aortic gradient independently predicted maternal hospitalization and, along with severe AS, was associated with adverse fetal outcomes such as low birth weight and small for gestational age [24]. These findings highlight the importance of careful management and preconception counseling.

Childbirth remains high risk in AS, even for those asymptomatic during pregnancy, as delivery and postpartum hemodynamic stress can precipitate symptoms. A multidisciplinary team approach is essential. Vaginal delivery is generally preferred due to smaller blood volume shifts, reduced bleeding, and avoidance of surgical complications—especially in patients with hypertrophied left ventricles sensitive to rapid preload and afterload changes. Early epidural analgesia is often advised to reduce cardiac stress, while cesarean delivery is reserved for obstetric indications, significant aortic dilation (>45 mm), prior dissection, or overt heart failure [1,24]. Individualizing the delivery plan helps optimize maternal and fetal outcomes while minimizing AS-related risks. In our case, intrauterine growth restriction combined with a dilated ascending aorta prompted a cesarean section with specific anesthetic measures. Impaired fetal growth in severe AS may result from hemodynamic compromise and reduced utero-placental blood flow. This was supported in our case, where intrauterine growth restriction occurred despite maternal gestational diabetes—a rare combination.

In our case, the patient’s stable condition and absence of clinical symptoms did not necessitate urgent advanced imaging of the aorta. The potential radiation exposure in a pregnant patient has traditionally posed challenges and concerns. However, a growing body of evidence suggests that modified CT scans performed after fetal organogenesis carry minimal risk when used appropriately. MRI is also a viable alternative to CT. Efforts to minimize radiation exposure to the fetus, such as minimizing fluoroscopy time and the use of sub-diaphragm lead shielding, can be used [19,25].

Managing a pregnant woman with severe aortic stenosis necessitates careful planning and discussion within a multidisciplinary team, as well as with the patient. A collaborative, multidisciplinary approach is essential to optimize both maternal and fetal outcomes. A key aspect of this management is preparing for potential cardiac decompensation, with four primary approaches, as pointed out before [14]:(1)Medical management until fetal viability, followed by early delivery.(2)Aortic balloon valvuloplasty as a bridge to term pregnancy, though this carries risks such as stroke, suboptimal relief, or severe regurgitation that may necessitate urgent transcatheter aortic valve implantation (TAVI).(3)TAVI during pregnancy, which in our case posed the potential risk of emergent conversion to surgical valve replacement due to the patient’s bicuspid aortic valve. Data from the literature also suggest some technical particularities that may occur in young women, such as femoral arteries with a small diameter and the absence of calcification on the aortic valve annulus, which could make anchoring of the prosthetic valve difficult [20].(4)Planned surgical valve replacement, which involves a high risk of fetal loss due to the need for cardiopulmonary bypass. The trigger for intervention should always be guided by the patient’s symptomatology during pregnancy [24].

Asymptomatic patients with normal LVEF are followed without treatment.

A thorough discussion with the patient is essential to weigh the risks and benefits of any intervention, including radiation exposure, the choice of aortic valve prosthesis implanted during pregnancy, and the potential need for anticoagulation. Each of these factors, along with their possible effects on the fetus, must be carefully considered, discussed and explained to the patient, in order to guide optimal decision-making.

Several interesting cases of pregnant women with AS are reported in the literature. Quain et al. described performing the Ross procedure in a pregnant woman with severe symptomatic aortic stenosis. The patient wanted to avoid long-term anticoagulation and had concerns about the limited durability of bioprosthetic valves [14]. Dawson et al. performed a successful aortic balloon valvuloplasty for symptomatic severe aortic stenosis in pregnancy [19]. In a landmark case, reported by Hoover et al., a patient with severe stenosis caused by degeneration of a bioprosthetic aortic valve underwent a successful valve-in-valve transcatheter aortic valve replacement (TAVR) at 29 weeks of pregnancy—marking the first reported use of TAVR during the third trimester [26].

A study by Galian-Gay et al. showed that pregnant women with BAV experienced few cardiac complications, with no cases of aortic dissection or surgery required. Although a modest yet significant increase in aortic size was noted during pregnancy, the overall risk of aortic complications remains low for those with baseline aortic diameters under 45 mm, though continued follow-up is advised [27].

Regarding the pregnancy outcomes for women with thoracic aortic disease, these are generally favorable when managed by current guidelines. While beta-blockers are routinely prescribed to reduce aortic complications, evidence supporting their utility remains limited, and their effects on fetal growth require further study. Atenolol, in particular, is linked to higher risks of fetal growth restriction, making alternatives like metoprolol, labetalol, or celiprolol (for vascular Ehlers–Danlos syndrome) safer choices, with fetal monitoring remaining essential [1].

Prophylactic aortic surgery thresholds before conception lack consensus, but European guidelines advise against pregnancy in Marfan syndrome patients with aortic roots >45 mm. Below this threshold, dissection rates are low, though family history of dissection or rapid aortic growth (>3 mm/year) significantly elevate risk and must be considered during the pre-pregnancy counseling [1]. Reassuringly, data from the ROPAC registry report low aortic dissection rates and positive outcomes in these patients [1,28].

A detailed analysis of pregnant patients with bicuspid aortic valves by Yuan et al. revealed comparable outcomes between syndromic and non-syndromic groups. Maternal mortality was reported at 50% in the syndromic group compared to 28.6% in the non-syndromic group (*p* = 0.4959), and fetal mortality was 25% versus 0%, respectively (*p* = 0.1987). The study noted that, as pregnancy advanced through the trimesters, both the peak and mean pressure gradients across the aortic valve increased significantly, indicating worsening stenosis. Notably, there was a marked reduction in the peak pressure gradients postpartum. Furthermore, the calculated aortic valve area showed a significant decrease during the third trimester compared to pre-pregnancy values, with this reduction persisting into the postpartum period [29]. These findings suggest that the hemodynamic burden of aortic stenosis may escalate during pregnancy and partially improve after delivery, which highlights the need for increased surveillance for possible cardiovascular decompensation during the last trimester of pregnancy.

The development of postpartum pre-eclampsia in our case significantly increased the risk of maternal morbidity, including the risk of aortic dissection, thereby further complicating the scenario. Despite extensive research into preventive measures, the incidence of pre-eclampsia has remained relatively stable over the past few decades. This persistence is likely due to the incomplete understanding of its underlying pathophysiology. Recent evidence suggests that suboptimal trophoblastic invasion disrupts the balance between angiogenic and antiangiogenic proteins, leading to widespread inflammation, endothelial damage, increased platelet aggregation, and thrombotic events accompanied by placental infarcts. Aspirin, when administered at doses below 300 mg, selectively and irreversibly inhibits the cyclooxygenase-1 enzyme, reducing prostaglandin and thromboxane production, and subsequently decreasing inflammation and platelet aggregation. This mechanism has sparked the hypothesis that aspirin may be beneficial in preventing pre-eclampsia [10]. However, despite receiving low-dose aspirin from the first trimester, our patient still developed pre-eclampsia, indicating that this preventive measure was not effective in her case.

Furthermore, the hyperglycosylated form of human chorionic gonadotropin (H-hCG) plays a crucial role in facilitating cytotrophoblast invasion, which is vital for successful pregnancy implantation and placental development. Impaired cytotrophoblast invasion can lead to inadequate transformation of spiral arteries, resulting in defective placental perfusion. This compromises the fetus’s nutrient and oxygen supply, leading to growth restriction as an adaptive response to the adverse intrauterine environment. Elucidating the function of H-hCG in placental evolution may reveal new strategies for managing and treating fetal growth restriction, as well as preventing pre-eclampsia [30].

Postpartum pre-eclampsia is a frequently underdiagnosed condition that can arise after a pregnancy, even in the absence of a prior hypertensive disorder diagnosis. It can also occur following pregnancies complicated by gestational hypertension or in women with underlying chronic hypertension [11]. Our case underscores the importance of heightened awareness for this pathology, as timely recognition and management are crucial to prevent severe maternal morbidity.

BAV follows an autosomal dominant inheritance pattern but shows incomplete penetrance and variable clinical severity, reflecting its complex genetic basis involving multiple interacting genes [31]. Notably, haploinsufficiency of the NOTCH1 gene is associated with BAV development [32,33]. First-degree relatives of individuals with BAV have a significantly elevated risk, with a prevalence rate 10 times higher than that observed in the general population [3,34]. While most individuals with BAV lack syndromic features, they often exhibit a spectrum of congenital cardiovascular abnormalities with variable severity. Specifically, BAV is strongly associated with left-sided cardiac lesions, including coarctation of the aorta (CoA) in 7% of cases, patent ductus arteriosus (8.5%), mitral valve anomalies (11%), ventricular septal defects (14%)**,** and thoracic aortic aneurysms in up to 50% of patients [7,35]. Comprehensive cardiovascular evaluation should always be performed during BAV management in order to identify associated comorbidities in a timely manner.

Those malformations can be present alongside sub-valvular, valvular, or supravalvular aortic stenosis, as well as mitral valve abnormalities, including mitral valve stenosis [36]. Research also indicates that the coexistence of BAV and CoA is more strongly associated with aortic dilatation than either condition alone. Structural changes in the aortic wall, affecting both the ascending and descending aorta, have been observed in CoA patients, contributing to increased stiffness in the aorta and carotid arteries [37,38,39]. In our patient, the aortic CT excluded the co-existence of CoA after pregnancy. According to the guidelines [40], a fetal cardiac malformation specialist performed serial detailed morphological scans of the fetus in order to exclude an inherited disease.

Pre-excitation is determined by the conduction properties of the atrioventricular (AV) node, which can be influenced by factors such as autonomic tone, medications, conduction system disorders, or mechanical trauma [41]. Accessory pathway unmasking can be carried out by an adenosine provocation test, but it has also been observed during anesthesia [42], during His bundle ablation procedures [43], and following tricuspid valve replacement [44].

The incidence of newly discovered pre-excitation after surgical or transcatheter aortic valve replacement is still to be determined, but there are few case reports of the unmasking of an AP after aortic valve replacement [41,45,46]. Among the proposed mechanisms, impaired AV conduction is believed to be the primary factor that facilitates accessory pathway conduction. Perioperative or post-operative conduction abnormalities are most often due to trauma affecting the AV node, His bundle, or the left and right bundle branches, given the close proximity of the conduction system to the aortic valve. Additionally, post-operative inflammatory responses can further contribute to these conduction disturbances. Beyond these factors, the use of dromotropic negative medications may further compromise conduction [12,41]. Furthermore, surgery could induce lesions to the atrioventricular conduction system and lead to atrioventricular block [47]. In our patient, the cardiopulmonary bypass time was 93 min, the aortic cross-clamp time was 69 min, and the estimated blood loss was approximately 500 mL—all within the expected range for this type of combined procedure. We believe that these parameters, being non-excessive, make it unlikely that conduction disturbances were caused solely by prolonged ischemia or intraoperative bleeding. Rather, the surgery appears to have unmasked a pre-existing accessory pathway, which had remained clinically silent prior to the intervention. The temporary absence of ventricular pre-excitation immediately after surgery may have originated from transient myocardial edema and injury, likely suppressing the excitability and conduction capacity of the accessory pathway. As cardiac function recovered and anti-arrhythmic medications took effect, the characteristic ventricular pre-excitation pattern reemerged, aligning with expected electrophysiological behavior.

The peculiarity of this case lies in the intriguing combination of pregnancy in a woman with a bicuspid aortic valve, severe aortic stenosis, and ascending aorta dilation, all of which remained asymptomatic throughout the pregnancy. However, complications arose in the form of intrauterine growth restriction, postpartum pre-eclampsia, and the discovery of an accessory pathway following aortic valve replacement. Despite these challenges, all complications were successfully managed. This case underscores the potential for both maternal and fetal complications in pregnancies involving mothers with aortic valve pathologies and ascending aorta dilation, emphasizing the importance of close monitoring by a multidisciplinary team.

## 4. Conclusions

Instances of severe aortic stenosis during pregnancy are rare, presenting a dual challenge due to limited experience and high-risk implications for both mother and baby. Hemodynamic fluctuations can lead to falsely elevated transvalvular aortic gradients, necessitating regular echocardiography and electrocardiography assessments throughout pregnancy. Delivery requires meticulous anesthetic and obstetric attention to mitigate the risks of hypovolemia, sudden preload increases, vasodilation, and arrhythmias. Despite the severity of the patient’s condition, she remained asymptomatic before conception and maintained good functional status throughout pregnancy and the peripartum period. Successful aortic valve replacement and ascending aorta reduction plasty were performed six months postpartum, revealing an accessory electrical pathway previously unidentified during the post-surgical phase. Given the potential hereditary nature of cardiac malformations, fetal echo-graphic screening was conducted by the obstetrician to rule out inherited conditions.

Advancements in medical care enable more women with heart disease to enter their childbearing years. The coordination of care should be done by a Pregnancy Heart Team comprising cardiologists, obstetricians, geneticists, and anesthesiologists, ensuring comprehensive and specialized management tailored to individual patient needs, with regular reassessment for possible intervention, taking into account changing maternal, obstetric, and fetal concerns.

## Figures and Tables

**Figure 1 diagnostics-15-02099-f001:**
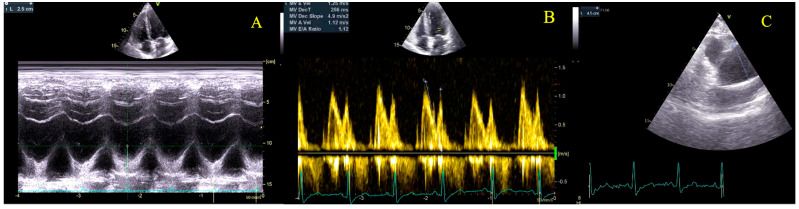
Transthoracic Echocardiography at 22 weeks of gestation. Panel (**A**)—Tricuspid Annular Plane Systolic Excursion (TAPSE) measurement; Panel (**B**)—Mitral inflow; Panel (**C**)—Ascending aorta measurement.

**Figure 2 diagnostics-15-02099-f002:**
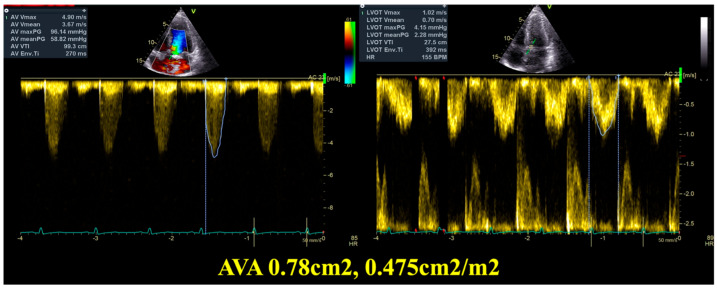
Transthoracic Echocardiography at 22 weeks of gestation—hemodynamic parameters and calculation of the aortic valve area (AVA) through the continuity equation. (**Left**)—Measurements at the level of the Aortic valve (VTI—velocity time index, maximum and mean gradient, maximum and mean velocity); (**Right**)—Measurements at the level of the left ventricle outflow tract (LVOT) (VTI—velocity time index, maximum and mean gradient, maximum and mean velocity).

**Figure 3 diagnostics-15-02099-f003:**
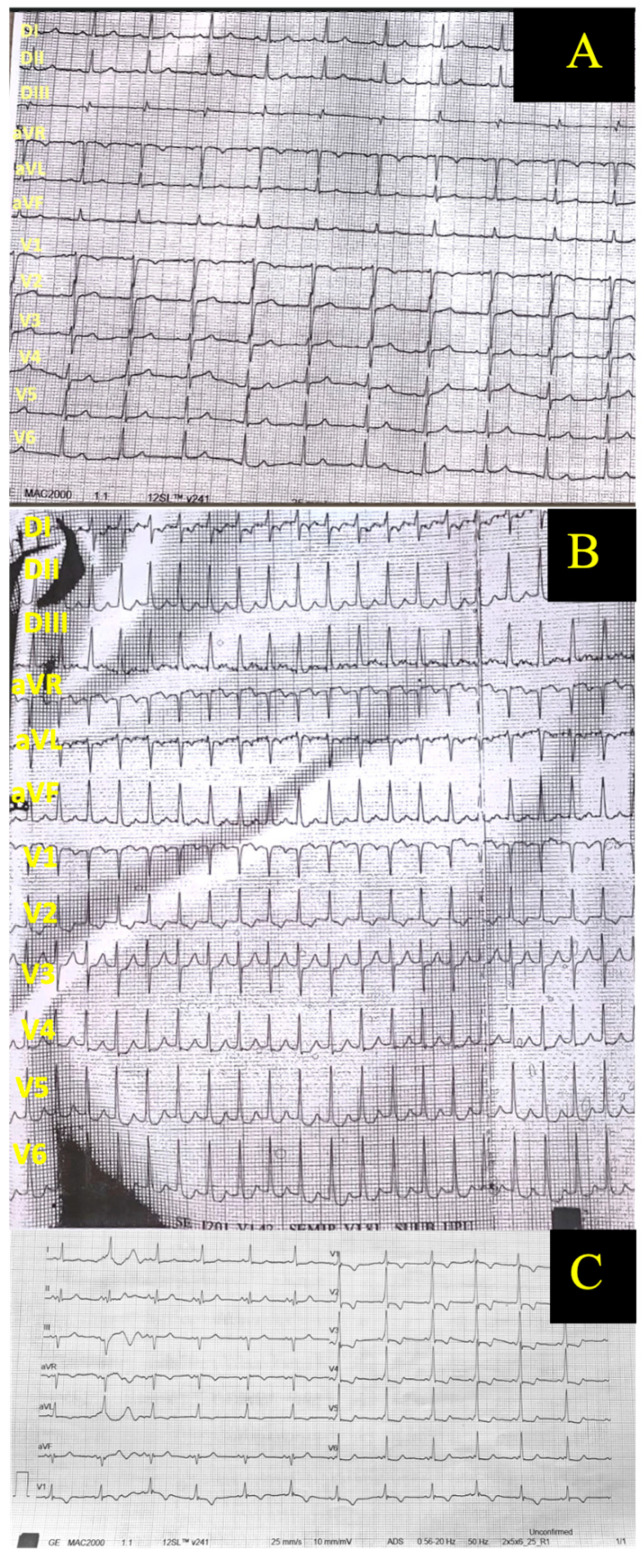
Electrocardiograms. Panel (**A**)—At baseline, before heart surgery; Panel (**B**)—During atrio-ventricular re-entrant tachycardia; Panel (**C**)—After conversion to sinus rhythm.

**Figure 4 diagnostics-15-02099-f004:**
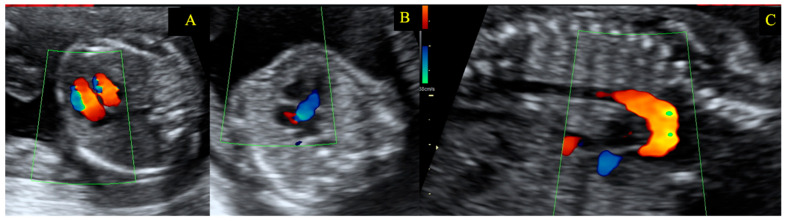
Fetal echography showing no cardiac or aortic malformation. Panel (**A**)—Normal filling of the heart chambers; Panel (**B**)—Normal origin of the aorta, no stenosis; Panel (**C**)—Sagittal view showing a normal aorta, no coarctation.

**Figure 5 diagnostics-15-02099-f005:**
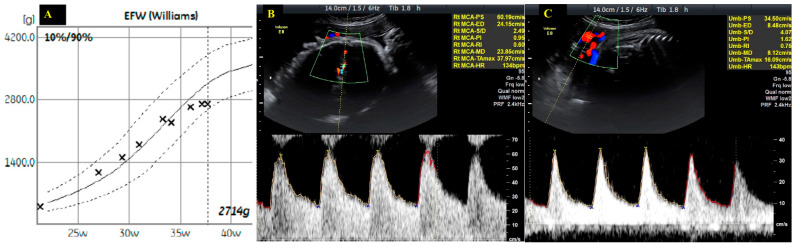
Echo-graphic signs of intrauterine growth restriction. Panel (**A**)—Growth chart showing the flattening of the fetal growth curves; Panel (**B**)—Doppler ultrasound showing decreased resistance and pulsatility in the middle cerebral artery; Panel (**C**)—Doppler ultrasound showing increased resistance and pulsatility in the umbilical artery.

**Figure 6 diagnostics-15-02099-f006:**
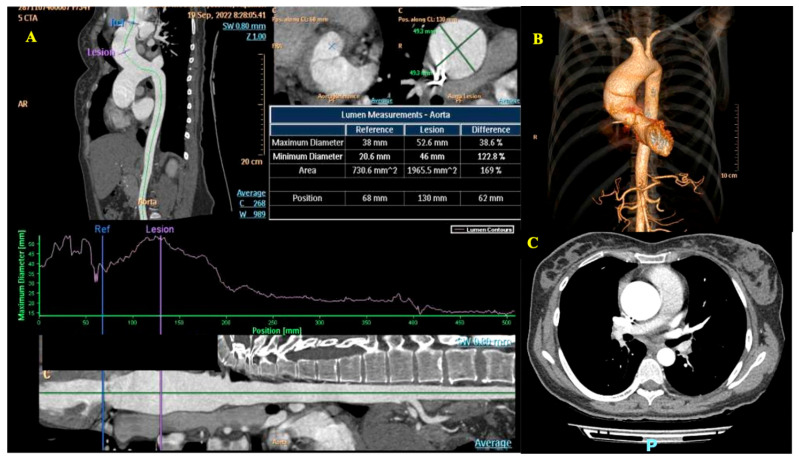
Aortic computed tomography. Panel (**A**)—Measurements of the aortic lumen; Panel (**B**)—Digital reconstruction of the aorta; Panel (**C**)—Transverse section at the level of the ascending aorta.

**Figure 7 diagnostics-15-02099-f007:**
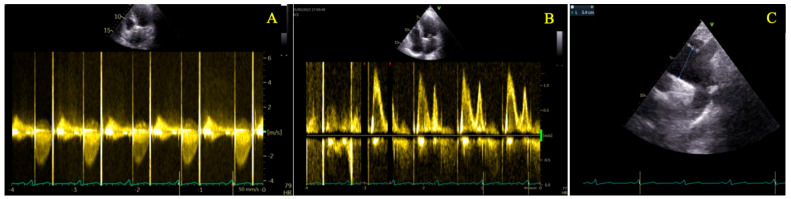
Echocardiography after surgery. Panel (**A**)—Aortic valve velocities; Panel (**B**)—Mitral valve inflow; Panel (**C**)—Ascending aorta measurement.

**Table 1 diagnostics-15-02099-t001:** The evolution of clinical, biological and echocardiographic parameters of the patient throughout the pregnancy.

Gestational Age (Weeks)	Heart Rate (bpm)	Hb (g/dL)	NT-proBNP (pg/mL)	6mWT (m)	LVEF (%)	LVOT VTI (cm)	AV VTI (cm)	Maximum AV Gradient (mmHg)	Mean AV Gradient (mmHg)	Indexed Aortic Valve Area (cm^2^/m^2^)
22	76	11.5	45	670	59	27.5	99.3	96	58	0.475
26	73				60	27.3	99.2	94	60	0.475
29	82	11.4	59	625	60	26.9	99.8	93	60	0.463
30	85				61	27	102	108	60	0.457
31	81				60	26.9	101.8	95	58	0.469
32	78	12.1	67	630	59	27.2	100	103	62	0.469
33	76				59	27.5	99.5	100	61	0.475
34	84				61	27.9	101	101	60	0.475
35	82	12.5	37	615	61	28	100.6	95	55	0.481
36	80				60	26.4	99	89	52	0.463
37	85	12.6			62	27	100	93	56	0.469
38	86	12.3	66	575	62	26.5	99.8	92	55	0.457
3 months postpartum	70	11.9	54	654	60	28.3	118	103	68	0.50

pm—beats per minute, Hb—hemoglobin, NT-proBNP—N-terminal pro b-type natriuretic peptide, 6mWT—6 min walking test, LVOT—left ventricle outflow tract, VTI—velocity time index, AV—aortic valve.

## Data Availability

Dataset available on request from the authors.

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
