# Peer review of "Severe Aortic Stenosis and Pre-Excitation Syndrome in Pregnancy—A Multidisciplinary Approach"

_diagnostics, 2025, doi:10.3390/diagnostics15162099_

Round 1

Reviewer 1 Report

Comments and Suggestions for Authors

The authors present the case report on the follow-up of pregnancy and labor in a woman with severe aortic stenosis.

The case is well presented, and exceptionally well-illustrated.

References are relevant to the research and include 14 items belonging to the recent 5 years (47 references in total).

The only problem with this article is very extended discussion.

I suggest to shorten fragments: lines 253-273; 306-324; 325-334.

Fragment 375-398 may be moved to the beginning of the chapter; as well as fragment 448-460.

Author Response

The authors present the case report on the follow-up of pregnancy and labor in a woman with severe aortic stenosis.

The case is well presented, and exceptionally well-illustrated.

References are relevant to the research and include 14 items belonging to the recent 5 years (47 references in total).

The only problem with this article is very extended discussion.

I suggest to shorten fragments: lines 253-273; 306-324; 325-334.

Fragment 375-398 may be moved to the beginning of the chapter; as well as fragment 448-460.

Response:

We sincerely thank the reviewer for the thorough and positive evaluation of our work.
We appreciate your kind remarks regarding the presentation, illustrations, and relevance of the references.
We agree with your observation that the discussion section is rather extended. We will revise and condense it to ensure it remains focused, concise, and directly aligned with the main findings of the case report.

We compressed the lines 253-273 into the following paragraph:

“Managing pregnancy in women with severe aortic stenosis is challenging due to complex clinical presentations. As noted in the literature, the differential diagnosis for heart failure symptoms in pregnancy includes physiological changes, cardiomyopathies, endocarditis, myocarditis, pulmonary embolism, myocardial infarction, spontaneous coronary artery dissection, arrhythmias, and pulmonary hypertension [14]. Pregnancy induces marked cardiovascular changes: cardiac output rises 30–50%, initially due to a from a 40% increase in plasma volume and later due to a 15–20% rise in heart rate, before declining slightly in late pregnancy due to uterine compression. During labor, cardiac output rises further 15–20% early, 50% in active labor, and up to 80% postpartum—driven by pain, contractions, and autotransfusion after delivery [15–17]. Systemic vascular resistance and mean arterial pressure fall early due to placental circulation and vasodilators. Relative plasma volume expansion leads to physiologic anemia, reducing viscosity and improving placental perfusion. Most changes resolve within two weeks postpartum, with complete recovery in about six months, but they can significantly increase risks in women with cardiac disease [18].”

Then, we abreviated lines 306-324 and 325-334 into:

“Heart failure affected 6.7% of previously asymptomatic patients but rose to 26.3% in symptomatic women with severe AS. Cesarean section rates were higher in severe AS (75.0%) versus moderate AS (48.3%), mostly for maternal indications (65.9%). Neonates of mothers with severe AS had lower median birth weights (3000 g vs. 3200 g) and shorter gestation (37.9 vs. 39.0 weeks). Peak aortic gradient independently predicted maternal hospitalization and, along with severe AS, was associated with adverse fetal outcomes such as low birth weight and small for gestational age [24]. These findings highlight the importance of careful management and preconception counseling.
Childbirth remains high risk in AS, even for those asymptomatic during pregnancy, as delivery and postpartum hemodynamic stress can precipitate symptoms. A multidisciplinary team approach is essential. Vaginal delivery is generally preferred due to smaller blood volume shifts, reduced bleeding, and avoidance of surgical complications—especially in patients with hypertrophied left ventricles sensitive to rapid preload and afterload changes. Early epidural analgesia is often advised to reduce cardiac stress, while cesarean delivery is reserved for obstetric indications, significant aortic dilation (>45 mm), prior dissection, or overt heart failure [1,24]. Individualizing the delivery plan helps optimize maternal and fetal outcomes while minimizing AS-related risks. In our case, intrauterine growth restriction combined with a dilated ascending aorta prompted a cesarean section with specific anesthetic measures. Impaired fetal growth in severe AS may result from hemodynamic compromise and reduced utero-placental blood flow. This was supported in our case, where intrauterine growth restriction occurred despite maternal gestational diabetes—a rare combination.”

We also shortened lines 377-385.

  • Fragment 375-398 may be moved to the beginning of the chapter; as well as fragment 448-460.

Response:

We thank the reviewer for the thoughtful suggestion regarding relocating fragments 375–398 and 448–460 to the beginning of the chapter. After careful consideration and extensive discussion among all authors, we decided to retain the original layout, as we believe it ensures better narrative flow and maintains the logical progression of the text.

Reviewer 2 Report

Comments and Suggestions for Authors

How did severe aortic stenosis and dilation of the ascending aorta lead to
a delay in intrauterine development?

Line 193 - what kind of decompensation of cardiac activity are we talking about?
 Line 225 - what operation was performed? prosthetics with valve-containing conduit or separate prosthetics of the aortic valve with supracoronor prosthetics of the ascending aorta? Why wasn't the choice made in favor of a biological valve or conduit, taking into account future pregnancy planning? 
What was the blood loss, the time of artificial blood circulation, as all this affects the impaired conduction of electrical impulses in the myocardium.

Line 246 - why were there no attempts to medically suppress additional pathways at first?

In general, the article is not unique, it is written in detail, and may be of interest to both specialists involved in pregnancy management, as well as cardiothoracic surgeons and cardiologists.
The main point is as follows - in my opinion, the statement about the effect of aortic valve stenosis on fetal development delay sounds unreasonable, especially at such a late stage, without substantiating decompensation of cardiac activity, or prove the opposite.

Author Response

  • How did severe aortic stenosis and dilation of the ascending aorta lead to
    a delay in intrauterine development?

Response:

Severe aortic stenosis with ascending aorta dilation can contribute to intrauterine growth restriction (IUGR) primarily through impaired uteroplacental perfusion. The two most frequent causes of IUGR are (1) a fetal genetic condition and (2) placental hypoperfusion. In our case, the obstetrician differentiated between these by performing detailed Doppler ultrasound of the fetal and placental circulation, which confirmed abnormal flow patterns (cerebral vasodilatation – brain sparing, umbilical cord arteries vasoconstriction were clear signs of fetal distress, while the uterine arteries had normal doppler waveforms, meaning that trophoblast invation and uterine artery remodeling were normal and not the cause of placental hypoperfusion) consistent with uteroplacental hypoperfusion rather than a fetal genetic abnormality. Although our patient’s resting cardiac output and cardiac index were within the normal range, it is important to note that, in the context of severe aortic stenosis, these values may still be relatively reduced compared with those of a pregnant woman without significant outflow obstruction (the normograms for cardiac output are generally defined by non-pregnant women values). During pregnancy—particularly in the third trimester—the reduction in peripheral vascular resistance and plasma volume expansion are expected physiological adaptations that lead to a marked increase in cardiac output. In patients with severe aortic stenosis, the fixed outflow obstruction limits the ability to further augment stroke volume in response to these demands. This relative hemodynamic limitation, even in the absence of an overt low-output state at rest, can result in suboptimal placental perfusion and chronic fetal hypoxia, ultimately leading to growth restriction. This pathophysiological mechanism is also well described in the literature for similar cases, further supporting our conclusion.

We also clarified this in the manuscript by adding the following paragraph to lines 284-291: “In our case, the presence of a fixed outflow obstruction, limits the capacity to further augment stroke volume, which may impair uteroplacental perfusion even without an overt low-output state. This relative hemodynamic restriction can lead to chronic placental hypoxia and intrauterine growth restriction, as confirmed by Doppler ultrasound assessment (cerebral vasodilatation – brain sparing, umbilical cord arteries vasoconstriction were clear signs of fetal distress, while the uterine arteries had normal doppler waveforms, meaning that trophoblast invation and uterine artery remodeling were normal and not the cause of placental hypoperfusion).“

  • Line 193 - what kind of decompensation of cardiac activity are we talking about?

Response:

We thank the reviewer for highlighting the lack of specificity in the original wording. We have revised the text to read “cardiac function deterioration,” as this term more precisely reflects the potential decline in both left- and right-sided systolic and diastolic function. These impairments are among the most frequent complications in such cases and can occur gradually or acutely during pregnancy, delivery, or the postpartum period.

  • Line 225 - what operation was performed? prosthetics with valve-containing conduit or separate prosthetics of the aortic valve with supracoronor prosthetics of the ascending aorta? Why wasn't the choice made in favor of a biological valve or conduit, taking into account future pregnancy planning? 

Response:

We thank the reviewer for this very pertinent observation. Indeed, in the original version of the manuscript we did not provide a detailed description of the surgical procedure, and we agree that this information is important for clarity and completeness. As mentioned earlier, the surgery was performed six months postpartum, at which time the ascending aorta measured 49 mm in diameter. The procedure consisted of separate mechanical aortic valve replacement and supracoronary non-reinforced reduction ascending aortoplasty (RAA), in accordance with the technique described by Vistarini et al. (PMCID: PMC9357918). Briefly, a wedge-shaped segment of the ascending aorta was excised by performing a longitudinal aortotomy starting next to the aortic clamp at the greater curvature, followed by closure with a continuous 4-0/5-0 prolene suture in a double-layered fashion. As shown in the referenced study, non-reinforced RAA offers excellent short- and long-term outcomes in appropriately selected patients with borderline ascending aortic aneurysms, which was the case of our patient. After thorough discussion with the heart team and detailed informed consent, the patient chose a mechanical valve for its superior durability, as she was certain she did not wish to have any future pregnancies, and RAA instead of ascending aortic replacement with a prosthesis or conduit, given the favorable anatomy and the desire to avoid more extensive surgery.

The following paragraph was added to the manuscript in the lines 217-226: “Successful aortic valve replacement with a mechanical prosthesis, chosen considering both the patient’s age and her stated wish to avoid future pregnancies, and supracoronary non-reinforced reduction ascending aortoplasty (RAA) were performed via ministernotomy (with the cardiopulmonary bypass time of 93 minutes, the aortic cross-clamp time of 69 minutes, and the estimated blood loss of 500 mL). A wedge-shaped segment of the ascending aorta was excised through a longitudinal aortotomy starting next to the aortic clamp at the greater curvature, and the aortotomy was closed with a continuous suture in a double-layered fashion. The decision for a mechanical valve and RAA, rather than prosthetic replacement of the ascending aorta, was made in full agreement with the patient to ensure valve durability while avoiding more extensive surgery.”

  • What was the blood loss, the time of artificial blood circulation, as all this affects the impaired conduction of electrical impulses in the myocardium.

Response:

We thank the reviewer for this relevant question and agree that these intraoperative parameters may influence postoperative myocardial conduction. In our case, the total cardiopulmonary bypass time was 93 minutes, with an aortic cross-clamp time of 69 minutes. The estimated intraoperative blood loss was approximately 500 mL. We note that these values are within the expected range for this type of combined procedure performed via ministernotomy. While prolonged cardiopulmonary bypass and cross-clamp times, as well as significant intraoperative blood loss, have been associated with a higher risk of conduction disturbances, in this patient these parameters were not excessive. Moreover, this is exactly what we described in the manuscript—that the surgery likely unmasked an accessory pathway which, based on the patient’s postoperative ECG findings, was present beforehand but had remained clinically silent. This phenomenon has been reported in the literature and is most likely related to subtle changes in atrioventricular nodal conduction occurring after valve surgery.

We modified the discussions at lines 482-492 into: “In our patient, the cardiopulmonary bypass time was 93 minutes, the aortic cross-clamp time was 69 minutes, and the estimated blood loss was approximately 500 mL — all within the expected range for this type of combined procedure. We believe that these parameters, being non-excessive, make it unlikely that conduction disturbances were caused solely by prolonged ischemia or intraoperative bleeding. Rather, the surgery appears to have unmasked a pre-existing accessory pathway, which had remained clinically silent prior to the intervention. The temporary absence of ventricular pre-excitation immediately after surgery may have originated from transient myocardial edema and injury, likely suppressing the excitability and conduction capacity of the accessory pathway. As cardiac function recovered and anti-arrhythmic medications took effect, the characteristic ventricular pre-excitation pattern reemerged, aligning with expected electrophysiological behavior.”

  • Line 246 - why were there no attempts to medically suppress additional pathways at first?

Response:

Emergency electrical cardioversion remains the first-line treatment when restoration of sinus rhythm is expected to provide hemodynamic benefit. Postoperative atrial fibrillation (POAF), defined as new-onset AF in the immediate postoperative period, is a common complication with clinical impact, occurring in 30–50% of patients after cardiac surgery. In our patient, AF presented with a rapid ventricular rate and hemodynamic decompensation (dyspnea, systolic blood pressure 90 mmHg). In this setting, immediate electrical cardioversion was indicated in accordance with the ESC Atrial Fibrillation Guidelines. The patient had already received beta-blocker therapy prior to cardioversion.

We changed the paragraph into: “On postoperative day 3 the patient had an episode of atrial fibrillation with rapid ventricular response and hemodynamic decompensation (dyspnea, systolic blood pressure 90 mmHg), converted to sinus rhythm by external electric shock.”

  • In general, the article is not unique, it is written in detail, and may be of interest to both specialists involved in pregnancy management, as well as cardiothoracic surgeons and cardiologists.
    The main point is as follows - in my opinion, the statement about the effect of aortic valve stenosis on fetal development delay sounds unreasonable, especially at such a late stage, without substantiating decompensation of cardiac activity, or prove the opposite.

Response:

We thank the reviewer for this comment and for recognizing the potential interest of our manuscript for specialists in pregnancy management, cardiothoracic surgery, and cardiology. We agree that the relationship between aortic valve stenosis and intrauterine growth restriction (IUGR), particularly in late pregnancy, should be clearly substantiated. In our case, although the patient did not present with overt clinical or echocardiographic signs of cardiac decompensation and her cardiac output/index at rest remained within normal limits for non-pregnant women, Doppler ultrasound performed by the obstetrician demonstrated abnormal fetal and placental circulation consistent with uteroplacental hypoperfusion (cerebral vasodilatation – brain sparing, umbilical cord arteries vasoconstriction were clear signs of fetal distress, while the uterine arteries had normal doppler waveforms, meaning that trophoblast invation and uterine artery remodeling were normal and not the cause of placental hypoperfusion). Uteroplacental hypoperfusion is one of the two most frequent causes of IUGR, the other being a fetal genetic abnormality, which was excluded in our case. We acknowledge that even in the absence of resting low cardiac output, the fixed obstruction of severe aortic stenosis limits the ability to further augment stroke volume in response to the increased hemodynamic demands of late pregnancy (particularly the third trimester), which can relatively reduce effective uteroplacental blood flow compared to healthy pregnancies. This pathophysiological mechanism, described in multiple reports and registry data, supports our conclusion that severe aortic stenosis can contribute to fetal growth restriction even in the absence of clinically apparent maternal decompensation. We have clarified this explanation in the revised manuscript, as mentioned before.

Reviewer 3 Report

Comments and Suggestions for Authors

Thank you to the editor for the opportunity to review the revised manuscript.
Congratulations to the authors on successfully managing a challenging clinical situation.
From my professional perspective, I have only one question and one comment for the authors.

It is necessary to include the CPB and clamping times of the ascending aorta in the surgical procedure. These are data that are typically reported in such studies.

What do the authors mean by the term: "reduction of the ascending aorta"? According to EACTS guidelines, replacement of the ascending aorta is indicated when the aortic diameter is more than 45 mm if an aortic valve operation is performed (IIa). If an ascending aorta replacement was performed, it should be clearly described. If a different procedure was chosen, I recommend that the authors explain in the text why a non-standard approach was used.

After adding the requested information, I recommend proceeding with publication of the article.

Author Response

  • It is necessary to include the CPB and clamping times of the ascending aorta in the surgical procedure. These are data that are typically reported in such studies.

Response:  

We thank the reviewer for this valuable observation and agree that including intraoperative parameters such as cardiopulmonary bypass (CPB) time, aortic cross-clamp time, and estimated blood loss provides important context, as these values are typically reported in similar studies. We have therefore updated the surgical procedure description in the Case Description section to include this information. In our case, the CPB time was 93 minutes, the aortic cross-clamp time was 69 minutes, and the estimated intraoperative blood loss was approximately 500 mL. All of these values are well within the expected range for a combined mechanical aortic valve replacement and supracoronary non-reinforced reduction ascending aortoplasty performed via ministernotomy. We believe that reporting these details enhances the completeness of our case description.

  • What do the authors mean by the term: "reduction of the ascending aorta"? According to EACTS guidelines, replacement of the ascending aorta is indicated when the aortic diameter is more than 45 mm if an aortic valve operation is performed (IIa). If an ascending aorta replacement was performed, it should be clearly described. If a different procedure was chosen, I recommend that the authors explain in the text why a non-standard approach was used.

Response:

We thank the reviewer for this important comment and the opportunity to clarify our terminology. By “reduction of the ascending aorta,” we were referring to a supracoronary non-reinforced reduction ascending aortoplasty (RAA) rather than replacement of the ascending aorta. As noted by the reviewer, the 2021 ESC/EACTS guidelines indicate replacement of the ascending aorta when the diameter exceeds 45 mm in patients undergoing aortic valve surgery (Class IIa). In our case, the ascending aorta measured 49 mm; however, after discussion within the multidisciplinary heart team and with the patient, we opted for RAA instead of prosthetic replacement. The decision was guided by several factors: Patient preference – The patient gave informed consent and expressed a clear wish to avoid more extensive surgery when a less invasive but effective alternative was available. Anatomical suitability – The ascending aortic dilation was localized, with favorable wall quality for aortoplasty, making RAA technically feasible and safe. Durability evidence – As described in the literature (e.g., Vistarini et al., PMCID: PMC9357918), non-reinforced RAA can provide excellent short- and long-term outcomes in appropriately selected patients with borderline ascending aortic aneurysms.

The procedure consisted of separate mechanical aortic valve replacement and supracoronary non-reinforced RAA. A wedge-shaped segment of the ascending aorta was excised via a longitudinal aortotomy starting next to the aortic clamp at the greater curvature, and the aortotomy was closed with a continuous double-layer 4-0/5-0 prolene suture. This approach was chosen in full agreement with the patient, who had no desire for future pregnancies and opted for a mechanical prosthesis combined with aortoplasty instead of prosthetic replacement of the ascending aorta/conduit. We have clarified this point and the rationale for our surgical choice in the revised manuscript.

Round 2

Reviewer 2 Report

Comments and Suggestions for Authors

The authors have substantially revised the manuscript, revising and improving it based on all the comments and recommendations.
Recommended for publication

Reviewer 3 Report

Comments and Suggestions for Authors

The authors have answered all of my questions. I recommend publishing the work in its submitted form.